# Let’s Walk It: Mobility and the Perceived Quality of Life in Older Adults

**DOI:** 10.3390/ijerph182111515

**Published:** 2021-11-02

**Authors:** Ulrike Bechtold, Natalie Stauder, Martin Fieder

**Affiliations:** 1Institute of Technology Assessment, Austrian Academy of Sciences, 1030 Vienna, Austria; 2Department of Anthropology, University of Vienna, 1030 Vienna, Austria; a01268625@unet.univie.ac.at (N.S.); martin.fieder@univie.ac.at (M.F.)

**Keywords:** quality of life, older adults, mobility, movement, ageing, active assisted living, assistive technologies

## Abstract

European policy and the research and development landscape put forward a number of arguments in favor of implementing “Active Assisted Living” (AAL) for older adults: it will improve older adults’ quality of life, allow them to age in place, and keep costs for an ageing society down by exploiting new technology markets. The idea is that older adults who are supported by AAL and make use of assistive technologies will enjoy more freedom, autonomy, and mobility and also improved social integration and better communication. Yet, despite a history of more than 10 years of European research and development, the use of AAL applications is not as widespread as expected. To examine older adults’ attitudes to assistive technologies, we conducted a study in Vienna (Austria) between 2018 and 2020 questioning 245 older adults aged 61–93 years (M = 74.27 SD = 6.654) who lived at their private homes and had different support needs (ranging from “no support” to “everyday visit of social and/or care organizations”). The three goals of the study encompassed: (1) examination of their quality of life, (2) their attitudes and use of assistive technologies, and (3) the way they perceive their own and others’ life-course and getting older. AAL as a concept links “ageing in place” and “quality of life”. However, “mobility” is also of major importance here. In this paper, we aim to investigate the relation between the independent variables “Quality of life” and “Mobility” and their possible associations with the following dependent variables: cohabitation, social integration, self-rated health, sportive activities, locomotion, home well-being and safety, physical limitations, falls, and self-perception of their own ageing (compared to others). We calculated multivariate models regressing on these explaining and confounding variables. We found a positive correlation between mobility and quality of life. In detail, our results show a significant positive association between QOL and mobility regarding self-rated health and self-perception. Experiencing vertigo, walking difficulties, and balance problems significantly and negatively influence self-rated health and self-perception compared to others. Our findings can also be read as a clear message that there is a need to improve both health and the culture of ageing and to facilitate positive attitudes toward ageing as an efficient way to enhance the Quality of life of older adults.

## 1. Introduction

Ageing in place is often depicted as the most desirable way of getting older, and the idea therefore frames scientific and policy debate by primarily focusing on supportive environments [1,2,3,4]. One way to support older adults to stay in their homes as long as possible is “Active Assisted Living “(AAL), which at the same time is designed to improve older adults’ quality of life and establish and support European industries and Europe as a market place (http://www.aal-europe.eu/ accessed on 28 October 2021). The term AAL goes back to European and national initiatives in research funding and technology development and embraces technological devices and systems intended to help older adults in critical areas such as mobility, health, inclusion, and the performance of their daily activities of living e.g., [5]. Yet, despite a history of almost 10 years of European research, the use of AAL applications is often not as widespread as expected, so the support they are intended to offer often fails to reach its target groups [6,7,8,9,10,11]. In order to be able to successfully implement AAL in practice, questions relating to perceived ageing and the significance of quality of life or the “good life” in the urban environment (socially and individually) must not be left out. Urban agglomerations—including Vienna—are subject to constant growth, and therefore the effects of demographic change (in the sense of a constantly changing age structure) will become particularly evident here. 

However, notwithstanding AAL, a certain degree of mobility in itself is a prerequisite for ageing in place, enabling older adults to successfully keep in touch with their environmental and social surroundings [3,12,13,14]. Yet, mobility is also described as a consequence of “ageing in place” [15,16,17].

This paper presents and discusses results of a study which aimed to investigate the quality of life and attitudes to AAL among older adults in Vienna. The main aim of this paper is to examine in what ways QOL and mobility relate to each other. 

Several studies have found a link between different modes of mobility and Quality of life (QOL) in older adults [14,18,19,20]. However, the scientific literature uses and measures the term “mobility” in various ways. One approach focuses on declining physical performance of older adults [21,22,23], which impedes older adults navigating their environment and fulfilling their activities of daily living (ADL) successfully. Another way is to examine older adults’ paths outside their own homes, considering also potential infrastructural difficulties, accessibility or barriers to transport systems, and their potential implications for older adults’ wellbeing, autonomy, and self-sufficiency [18,24,25]. 

In this paper, we conceive of older adults as the mobile, deciding, and active entity: we apply an approach that puts older adults, their activities, and their self-assessed QOL at the center of attention [26], bearing in mind that “feeling in control” maintains and facilitates all sorts of activities. 

Our definition of mobility refers to mobility in a broader way and entails how older adults deal with their ADL, including indoor and outdoor mobility and necessary and unnecessary activities [27]. We also include whether they are (positively) projecting themselves into concrete future activities and embracing cognitive, psychosocial, physical, environmental, and financial determinants for mobility after Webber et al., 2010 [14]. We asked our participants if they practice regular sportive activities, a potential facilitator for mobility. Experienced falls or specific health deficits were also investigated.

Our definition of Quality of life (QOL) embraces the sum of living conditions an individual experiences. It is empirically measured along questions covering the concrete social and economic background (see Table 1 and Table 2) as well as the subjective dimension of QOL regarding living conditions, infrastructure, interactions with others, social participation and support (see Figure 1 and Figure 2). Most importantly, we deploy the rating of one’s overall Quality of life (oQOL) and more detailed ratings of health and wellbeing [28] to cover QOL. In assessing the overall rating of one’s QOL, “How would you rate your overall quality of life?” (see method section, Figure 1), we refer to Wang et al. pp.3 in [29] who state that such an overall measure captures older adults’ “(…) overall psychosocial situation” from their own perspective and provides a “better methodological homogeneity and comparability of the condition of groups across studies and countries” Wang et al. pp.3 in [29]. Hence, we investigated the relation between the independent variables “Quality of life” and “Mobility” and their possible associations with the following dependent variables: cohabitation, social integration, self-rated health, sportive activities, locomotion, home well-being and safety, physical limitations, falls, self-perception of their own ageing (compared to others), and determinants of mobility, referring to Webber, 2010 [14].

The study deploys a processual approach and does not assume that ageing begins at a certain number of lived years (age). The subjective perception of getting older over time and how older adults see themselves (compared to others) is included. Together with the consideration of possible future plans, a life-course approach is adopted [30].

## 2. Materials and Methods

### 2.1. The Questionnaire 

We conducted a comprehensive study containing a questionnaire filled out by 245 older adults (aged 61–93 years), predominantly living in their own private homes in Vienna, between 2018 and 2020. The study was conducted to investigate older adults’ attitudes to assistive technologies and, along with this, relations to their reported overall Quality of life (oQOL) in Vienna, Austria. The questionnaire divides into three parts: -Living conditions and socioeconomic environment, social integration and state of health (incl. experiences of falling and hospitalization), perception of living and built environment, mobility, and QOL.-Assessment of and experiences with technical aids and the willingness to undertake concrete changes in their life or living environment.-Assessment of how their ageing is perceived (continuum/breaks) in terms of self-image and compared with others. Survey of wishes and ideas concerning the process of getting older (including the perceived significance of AAL—active assisted living).

Quality of life (QOL) is defined as the sum of living conditions and individual perception of these. It is empirically measured along questions covering the concrete social and economic background as well as the subjective dimension of QOL regarding living conditions, infrastructure, interactions with others, social participation and support, as well as a rating of one’s overall Quality of life (oQOL) and more detailed ratings of health and wellbeing and see also [28]. 

### 2.2. Procedure and Sampling 

The questionnaire was thoroughly pre-tested and validated (*n* = 12) in order to ensure compressive, clear, and relevant questions. Another result of the pre-test was an added option for those older adults who do not want to talk about their own ageing. 

The questionnaire was anonymous and emphasized privacy in all questions (e.g., instead of postal codes we asked for the size of the place of residence). We kept no lists of persons or locations used in gathering the questionnaires at any time. Thorough information about the goal and the use of data was provided, as well as a clear introduction explaining how to fill out the questionnaire. 

Parallel to pre-testing the questionnaire, the authors contacted all major institutions and organizations in Vienna that deal with older adults. Several organizations allowed us to approach their members and acted as multipliers (numbers of questionnaires in brackets): Vienna Pensioners Clubs (100), Viennese social services (45), the acute gerontology department of the Viennese Hartmann Hospital of the Elisabeth Hospital group (50); Association “alters.kulturen,” Vienna (30), citizens of previous ITA participation projects, selected parishes in Vienna; Board of the Austrian Senior Expert Pool in Vienna (58), information day on care of the Vienna Chamber of Labour (30), Association “culture in Vienna,” Vienna (35), Seniors at WUK, Vienna (30), “Seniors Dance Club,” Vienna (15), Students of the University of Vienna, approaching their grandparents and neighbors (35), “Seniors Club,” Graz (20), Association “Vita Activa at the Karl Franzens University,” Graz (20), Parish in Lower Austria (5), individual older adults, as registered within the database of former participatory projects (5). 

Before filling out the questionnaire, most persons were personally instructed: of 478 questionnaires distributed, only 135 were distributed without a face-to-face explanation beforehand. These 135 persons received a group introduction or a written guide. Moreover, selected older adults were personally sent the questionnaire (including a written guide) via snail mail—these persons had previously participated in citizen activities and had consented to being contacted in connection with future research activities by the authors’ institutions. 

A total of 245 correctly filled out questionnaires were given/sent back and therefore made up the data base.

### 2.3. Conceptual Thoughts

The main aim of this paper is to see in what ways QOL and mobility relate to each other; how are QOL and mobility associated with self-rated health as well as with the demographic characteristics of the study participants? We introduce the variables we tested in terms of direct and indirect mobility and QoL in the following. We depict these relations in Figure 1. Further down, the related questions are briefly summarized. 

### 2.4. Mobility

We assume that the older adults’ factual and direct mobility is expressed in their ability to perform the daily activities of living without problems (mostly) themselves (Q27/Q40). Indirectly, we assume that the factors sportive activity (Q26), physical limitations (Q41), and the way they individually picture the determinants of mobility (Q63) (Webber, 2010) influence actual mobility. We aim to test how these indirect aspects (Q26, Q41, Q45, Q63) and the direct expression of mobility interact (Q27/Q40). 

### 2.5. Quality of Life

We also assume that the experienced QOL of the questioned older adults is expressed in their rating of the oQOL (Q24). Indirectly, we assume that the factors cohabitation (Q14, Q17), social integration (Q21, Q23), perceived health (Q25), and the way they individually experience safety and wellbeing at home (Q38A) influence actual QOL. We aimed to test how these indirect aspects (Q14, Q17, Q21, Q23, Q25, Q38A) and the direct expression of QOL interact (Q24).

Moreover, we investigate whether the questioned persons have experienced falls (Q45), which could have an impact on both QOL and mobility. Additionally, we want to see how their primary form of locomotion (Q28), their self-assessed psycho-physical comparison with other people of the same age (Q61), as well as their individual approach to their ageing and the connected relativity of calendric age (Q2, Q62) affect their mobility patterns and their oQOL.

As both self-reported health and self-perception compared to others (a summary indicator of rated health, fitness, activeness, and being content compared to others) encompasses a wide range of conditions, we used these two variables as dependent in the multivariate models.

For all analyses, we used SPSS IBM, version 27, R 4.0.5 (R Core Team 2020) (IBM, Armonk, NY, USA), libraries ordinal and psych. 

### 2.6. Multivariate Statistics

To examine if QOL and mobility in terms of daily activities are associated with the demographic characteristics of the participants, we used two separate multivariate general linear models regressing:

(i) How would you rate your quality of life and (ii) executing daily activities regressing on sex, age, education, self-rating of health, and self-perception compared to others on the basis of a binomial error structure. 

To examine which conditions are associated with self-rated health compared to others, we regressed self-rated health using separate ordinal models (R library ordinal, function clm), on (a) memory problems, (b) walking problems, (c) debility of sight, (d) hardness of hearing, vertigo, (e) balance problems, and (f) orientation problems, controlling for sex, age, and education.

In the case of self-perception, we applied a general linear model, regressing self-perception on (a) memory problems, (b) walking problems, (c) debility of sight, (d) hardness of hearing, vertigo, (e) balance problems, and (f) orientation problems, controlling for sex, age, and education.

Furthermore, using tetrachoric correlations we calculated the correlation matrix among walking problems, vertigo, balance problems, and orientation problems.

We also verified the validity of the variable “executing daily activities” with the variable “does someone take care of your daily chores?” These variables overlap to ~85%. Comparable (~81%) holds true for “How would you rate health—rating health bad and intermediate,” and if someone has fallen once or more often since the age of 55. 

## 3. Results

### 3.1. Descriptive Statistics

#### 3.1.1. The Sample Composition—Description of Main Variables

The sample consisted of 151 women (62.4%) and 91 (37.6%) men (*N* = 242). The average age of male participants was 74 years (M = 73.6 SD = 6.601 *N* = 73), while the average age of female participants was 75 years (M = 74.6 SD = 6.682 *N* = 126). In total, 95% of the respondents who filled out the questionnaire lived in their own private homes, while 5% said they lived in some sort of supported environment. 

A total of 15.6% (38) of participants completed compulsory school, 37.3% (91) completed vocational or commercial school, 15.6% (38) attained a high school diploma, 11.5% (28) completed some form of college-related education, and 20.1% (49) of participants attained a college degree (*N* = 244). More than half of all participants (59.5% or 116 persons) lived in an area with more than 1,000,000 inhabitants at the time of the study (*N* = 195).

In total, 38.9% (93) of the participants have a net monthly income of EUR 1000–2000 at their disposal, 24.3% (58) have a net income of EUR 2000–3000, 10% (24) have EUR 500–1000, 8.4% (20) EUR 3000–4000, 3.3% (8) EUR 4000–5000, another 3.3% (8) more than EUR 5000, 2.9% (7) less than EUR 500, and 8.8% (21) of participants did not want to answer this question (*N* = 239).

#### 3.1.2. Social Aspects

Approximately half of the respondents (51.6% or 126 persons) reported living with at least one other person (*N* = 244), while 38.2% (73) of participants were living in a cohabitation (*N* = 191).

When participants were asked how often they met their most important contact persons, the most frequently used answers were daily (43.1%), several times per week (29.3%), and several times per month (31%) (*N* = 239). 

The social connectedness in older persons’ neighborhoods was high: 74.7% (180) maintained an active relationship with one or more of their neighbors (*N* = 241).

#### 3.1.3. Mobility

More than two thirds of the older adults (70.6% or 173 persons) regularly exercised (at least several times a month) (*N* = 245).

The most frequently used forms of locomotion outside older adults’ houses were public transportation (58.6% or 140 persons), walking (46.4% or 111 persons), using a car (45.6 or 109 persons), using a bicycle (10.9% or 26 persons), other forms of locomotion (8.8% or 21 persons), and using a taxi (7.9% or 19 persons) (*N* = 239).

When it came to doing their daily purchases and errands (Direct mobility, as measured by question 27), 81.8% (198) of the participants reported doing their chores themselves, while 18.2% (44) reported that someone else made their purchases for them (*N* = 242).

Physical limitations as manifestations of reduced mobility performance are measured by the question “Have you had any experience with the following limitations?”, which showed that the most commonly reported physical limitations experienced by participants were walking difficulties (49.1%, 109 persons, *N* = 222), followed by vision impairments (42.1%, 90 persons, *N* = 214), hearing impairments (41.9%, 90 persons, *N* = 215), and memory difficulties (37.6%, 80 persons, *N* = 213). Less frequently experienced limitations were vertigo (28%, 58 persons, *N* = 207), balance problems (23.5%, 47 persons, *N* = 200), and orientation problems (10.9%, 22 persons, *N* = 202).

#### 3.1.4. Quality of Life 

In total, 82.9% (203) of the participants reported a good QOL, while 17.1% (42) reported an intermediate (or bad) QOL (*N* = 245). Additionally, 70.2% (172) of participants reported being satisfied with their health status, 15.1% (37) had moderate health status, and 14.7% (36) were dissatisfied (*N* = 245).

Health satisfaction is measured by question 25, and 70.2% (172) of participants reported being satisfied with their health status, 15.1% (37) had moderate health status, and 14.7% (36) were dissatisfied (*N* = 245).

The question “How comfortable do you feel in your home during the day?” showed that the overwhelming majority of older adults (92.6% or 224 persons) felt comfortable in their own home, with 5.8% (14) feeling neither comfortable nor uncomfortable, and 1.7% (4) feeling uncomfortable in their home (*N* = 242). 

#### 3.1.5. Self-Perception

Compared to others their age 51% feel healthier (*N* = 224), 53.9% fitter (*N* = 230), 62.4% more active (*N* = 232), and 62.9% more content (*N* = 226), while 8.6% (21) feel less healthy, 16.7% (41) less fit, 15.5% (38) less active, and 5.7% (14) less content. A total of 31.8% (78) felt they could not compare their health with others, 23.3% (57) could not compare their fitness, 16.7% (41) could not compare their activity, and 23.7% (58) could not compare their contentedness with others. For all four questions, respondents rated themselves higher than their peers compared to people the same age. 

Specifically, 55.8% feel healthier (*N* = 224), 57.4% fitter (*N* = 230), 65.9% more active (*N* = 232), and 68.1% more content (*N* = 226), while 9.4% (21) feel less healthy, 17.8% (41) less fit, 16.4% (38) less active, and 6.2% (14) less content. There was a high level of uncertainty in relation to this question. In total, 34.8% (78) felt they could not compare their health with others, 24.8% (57) could not compare their fitness, 17.7% (41) could not compare their activity, and 25.7% (58) could not compare their contentedness with others.

The perceived age (measured by the question “Perceived age is not always the same as calendrical age. How old do you feel?”) given by the participants was on average 65.62 years (SD= 9,912, *N* = 198). The perceived age on average was lower than the actual age (M = 74.27 years, SD = 6,654, *N* = 199).

Participants were asked if their thoughts about their future ever included at least one of the options (1) “possible illnesses and ailments”, (2) “possible falls”, (3) “possible lack of independence”, (4) “possible isolation”, (5) “possible lack of support”, (6) “possible poverty”; 73.3% (170) of the older adults stated that they included at least one of these options in their thoughts about their future (*N* = 232).

### 3.2. Inference Statistics

Figure 2 illustrates the main results—how the independent variables mobility and Quality of life interact with the selected parameters. 

#### 3.2.1. Association of Main Variables

Both independent variables, “Quality of life” and “Mobility”, are correlated at 0.51 *** (tetrachoric correlation). They are also significantly related to self-rated health and self-perception compared to others as displayed in Figure 2. Multivariate Models and Regression Models.

#### 3.2.2. Quality of Life

By the regression of oQOL on sex, age, education, self-rated health, and self-perception compared to others, we found that education, self-rated health, and self-perception are significantly positively associated with oQOL. Self-rated health, in particular, is significantly associated with oQOL as indicated by the ODDS ratios: rating health as satisfactory raises the probability of rating oQOL as good by more than eight times compared to individuals who rate their health as unsatisfactory (see Table 1). Rating one’s health as intermediate increases the probability of rating oQOL as good by more than four times. Education and self-perception compared to others have a comparable effect, as indicated by ODDS ratios: raising both for one step increases the probability of rating oQOL as good by 1.7 or 1.5 times, respectively (see Table 1). 

#### 3.2.3. Mobility 

We found a comparable pattern for mobility, that is the regression of being able to execute activities of daily living: again, education, self-rated health, and self-perception compared to others are significantly positively associated with being able to execute activities of daily living. Although self-perception shows the most significant association, the ODDS ratio again indicates substantial effects on being able to execute activities of daily living: rating health as intermediate increases the probability of being able to execute activities of daily living by more than two times and rating health as satisfactory by more than 4.5 times compared to rating health only as unsatisfactory (see Table 2). A higher level of education increases the probability of being able to execute activities of daily living by 1.4 times and self-perception by 1.6 times, hence both have a comparable effect (see Table 2). 

#### 3.2.4. Association of Self-Rated Health, Self-Perception with Certain Conditions

We found that to some extent, related conditions (walking problems, debility of eyesight, vertigo, balance problems, and orientation problems) are significantly negatively associated with both self-rated health (ordinal regression—see Table 3) and self-perception (general linear model based on a Gaussian error structure—see Table 4). Hence these conditions, in particular, influence health perception and the perception in comparison to others negatively. 

#### 3.2.5. Correlation of Conditions—“Related Conditions”

Furthermore, these four conditions, which are presumably to some extent related, are indeed significantly positively correlated, as indicated by the tetrachoric correlations; hence, they may represent a “related bundle” (correlation matrix Table 5). 

## 4. Discussion

Generally, our sample composition is in line with the findings of the Austrian interdisciplinary study on the oldest old [31] in the aspects of gender distribution, education, income, number of persons living in the household, and number of children. The age range of our participants was 61–93 years, compared to 80–85 years in Ruppe et al. [31], 95% of our participants still lived in their own homes compared to 87% in Ruppe et al. [31], and 28% were widowed compared to 51% in Ruppe et al. [31]. Our sample displays significantly higher self-reported health, 70% compared to 56% in Ruppe et al. [3,29], and a significantly lower subjective age (79% compared to 60% who felt younger and 15% compared to 30% who felt their subjective age was approximately equal to their chronological age). These divergent findings could be related to the significant difference in age distribution in the whole sample. 

In our study, we found that Quality of life and Mobility were significantly correlated. This association can also be found in the literature [19,20]. However, neither our sample nor the literature we reviewed offers a clear causal relation between mobility and oQOL in terms of which one causes the other to change. The literature offers different explanations for this unclear causality. For example, Ravulaparthy et al. [19] analyzed mobility in a social context and found a relation between social engagement outside the home and life satisfaction; the causality they suggest is that social isolation following a lack of social engagement leads to a decrease in life satisfaction. These findings are contrasted with a systematic review by Campbell et al. [32], who looked at mobility from a purely physical perspective and found that most physical activity interventions did not improve QOL; only when the intervention improved QOL did it also improve physical function. They suggest that physical exercise could either indirectly improve QOL by improving physical function and/or physical limitations or directly improve QOL in an unspecified way.

Moreover, QOL was significantly positively associated with self-rated health, self-perception, and education. Here, our results align with Maniscalco et al. who found a correlation between QOL and self-rated health in a sample of individuals aged 50 and older, while according to Mendoza-Núñez et al. [33], older adults’ self-perception of old age influences their health. Ruppe et al. [31] showed that older adults’ socioeconomic status (including education) is strongly associated with their participants’ QOL. Although we initially expected an association between QOL and social integration, we found no association in the multivariate analysis. This lack of connection is not consistent with a body of literature on this topic that has found the QOL and older adults’ social life to be closely associated e.g., [34,35]. A possible explanation for this could be that our sample’s strongest predictor of a QOL was self-rated health. Without this variable, the social integration of older adults might have played a more significant role, leaving another large predictor (self-rated health) unaccounted for. Another possibility is the variety of ways in which researchers measure QOL. It is not used or defined consistently in the literature, meaning that the results of other studies might not be fully comparable to ours [32,36]. Moreover, the overall socioeconomic status in our sample is comparatively high, which could be contributing to the high self-rated health status.

For mobility, we found a similar pattern in our sample. It is significantly positively associated with self-perception, self-rated health, and age. These findings are consistent with Mifsud et al. [37] and Levy and Myers [38], who also found an association between self-perception and mobility. According to the literature, health and age have a major, if not the most important, impact on older adults’ mobility e.g., [31,39,40,41].

The tetrachoric correlation matrix indicates that walking problems, vertigo, balance problems, and orientation problems are correlated with each other and thus point to a mutual relatedness of these conditions. Vertigo has been associated with a range of illnesses, especially cognitive and psychiatric ones e.g., [42,43]. Our results and studies investigating balance problems and dizziness point to increased falls in affected older adults. Falls are known to increase the fear of falling (FOF) and to cause injuries and possible loss of independence, which in turn affect the QOL as well as mobility e.g., [44,45,46,47,48].

The literature frequently discusses the link between exercises and functional mobility e.g., [49]. Gómez-Bruton et al. [50] found that aerobic endurance, measured as meters covered in a six-minute walk, is a strong predictor of future health-related QoL in older adults. However, even though Langlois et al. [51] show that physical exercise at least three times a week over three months improves older adults’ QOL and their physical and cognitive performance, Campbell et al. [32] found in a systematic review of 15 published studies that the correlation between exercise, QOL, and ADL in frail older adults was inconsistent. Only studies that showed improvement in QOL and ADL also reported increased physical improvement in participants. Levy and Myers [38] show that preventive health behavior was influenced by participants’ subjective age and attitude toward ageing. It is possible that the social integration of older adults could have been found to play a significant role if we had investigated a larger number of cases. This may also hold true for other variables. In our case, at any rate, self-rated health plays the most crucial role. 

Another aspect that connects mobility and QOL for older adults is the “ageing in place” paradigm, which is widely accepted as the most desirable way of getting older e.g., [2,4]. Mobility is seen as a precondition and consequence of ageing in place e.g., [15,16,17]. Hence public authorities promote science, research, and technology development in order to elaborate supportive environments (e.g., AAL—ambient assisted living). This is a significant development and means that older adults are depicted in a dependent way, which could be a slippery slope or, at times, even slightly reductionist e.g., [52,53,54,55,56,57,58,59]. Autonomy seems to play an increasing role. For example, Ravulaparthy et al. [19] show that happiness and satisfaction levels in older age are lower in housing for the elderly and higher in retirement communities. Presumably the cause of this is easier social engagement between residents in retirement communities, which leads to more mobility and more happiness among residents. In total, 95% of persons we questioned lived in their own homes, and we still found no connection between social integration and QOL in our sample. We think this is because they display high health satisfaction. It could be essential that variables like social contacts depend on this and vice versa, so that they are mutually dependent on each other. However, based on our data, we are not able to figure out causalities.

## 5. Conclusions

We found significant correlations between mobility and oQOL. Yet, it remains unclear which is causing which. The literature provides different explanations, depending on what view is taken of this unclear causality. However, this may not be critical; perhaps the fact that they are connected allows us to focus on either of them and expect them to cause an improvement in the other. Acknowledging the apparent problem of not knowing which aspect is causally responsible for which consequence, socio-gerontological analyses could profit from the more in-depth analysis presented here. Primarily focusing on how older adults can use their surrounding world in a way that improves their QOL could be promising. This can be read as yet another proof that specific measures may not lead to the expected outcome, and ageing is more complex than sometimes assumed when unilinear causalities are expected. Nevertheless, our results confirm that presumably (all) measures that lead to better-perceived health will most likely support experienced quality of life and mobility. 

Vertigo, walking problems, and balance issues may need to be taken into the focus of assistive technologies in a more pronounced way. Other supportive strategies could also be promising. Medical approaches, care strategies, and senior societal organizations can, if they support mobility, contribute to higher QOL. 

Hearing and seeing deficiencies seem to have less influence on perceived QOL. Of course, for these many quite accessible solutions exist on the market, and they are characterized by a higher level of (societal) acceptance. 

Finally, the unquestioned persistence of ageing in place as the most desirable way of getting older, which continues to frame scientific and policy debate and primarily focuses on supportive environments (built spaces, smart housing, AAL technologies, etc.), needs to be questioned. It could easily mask the most crucial focus on older adults themselves and their perceived (and relative as highly individual) dynamic perceptions of ageing, ways of living, spaces, and places. 

However, keeping up older adults’ mobility is essential in many ways. Environments should support them or, to put it more radically, compensate for their declining abilities [54]. The widespread assumption is that supportive environments play a vital part in supporting older adults’ ability to keep up constant mobility patterns, and their implementation affects many actor groups: the policymakers who have to decide upon and finance supportive environments, researchers who design and develop these “prosthetic environments” [52,55], and the scientists who raise questions and seek answers, and lastly but surely most importantly the (re-)acting older adults themselves who actively use these environments e.g., [56]. 

Moreover, our findings show that a positive self-perception of older adults could contribute to improving oQOL. An improved societal attitude to ageing affects psychosocial aspects and the physical level, such as self-rated health and actual mobility. However, Levy and Myers [38] have already found an apparent effect of subjective age and attitude to ageing on the very tangible aspect of actually executing health behavior (exercises, etc.).

These results contain a clear call for decision-makers to actively support strategies that improve our culture of ageing and, along with this, a more positive and valuing attitude towards ageing, which can have tangible effects on older adults’ wellbeing.

## Figures and Tables

**Figure 1 ijerph-18-11515-f001:**
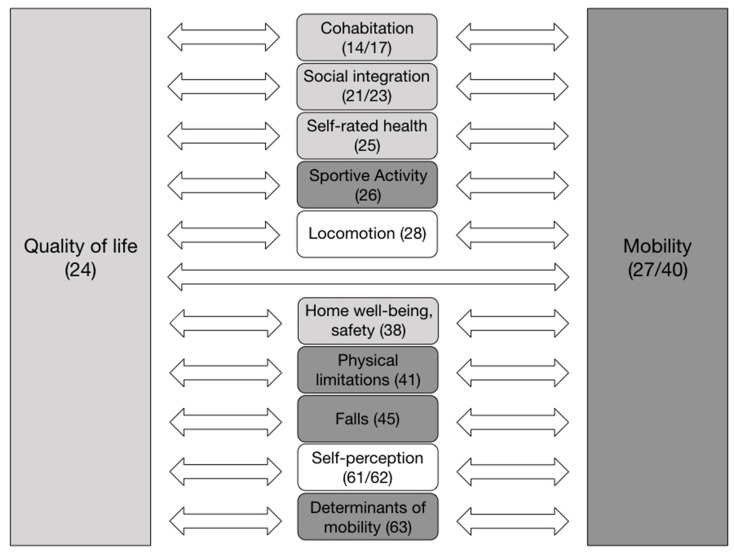
Representation of the independent variables “Quality of life” and “Mobility” and their possible associations with the dependent variables (the number of the relevant question is given in brackets).

**Figure 2 ijerph-18-11515-f002:**
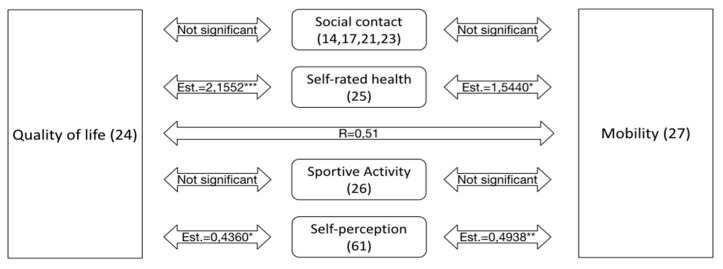
Interaction of the independent variables “Mobility” and “Quality of life” with the selected parameters. All four sub-questions of “Self-perception” are significant (the highest significance is indicated: * *p* < 0.05, ** *p* < 0.01, *** *p* < 0.001).

**Table 1 ijerph-18-11515-t001:** General linear model of oQOL regressing on sex, age, education, self-rated health, and self-perception compared to others based on a binomial error structure; * *p* < 0.05, *** *p* < 0.001.

	Estimate	Std. Error	z Values	OR	*p*	
(Intercept)	−1.8239	2.7647	−0.6600		0.5094	
Sex female (ref. male)	0.3235	0.5660	0.5720	1.3819	0.5676	
Age	−0.0089	0.0354	−0.2520	0.9911	0.8009	
Education	0.5522	0.2334	2.3660	1.7371	0.0180	*
Self-rated health intermediate (ref. unsatisfactory)	1.4695	0.6846	2.1460	4.3471	0.0318	*
Self-rated health satisfactory (ref. unsatisfactory)	2.1552	0.6135	3.5130	8.6296	0.0004	***
Self-perception compared to others	0.4360	0.1817	2.4000	1.5466	0.0164	*

**Table 2 ijerph-18-11515-t002:** General linear model of being able to execute activities of daily living regressing on sex, age, education, self-rated health, and self-perception compared to others on basis of a binomial error structure; * *p* < 0.05, ** *p* < 0.01.

	Estimate	Std. Error	z Values	OR	*p*	
(Intercept)	3.6667	2.8055	1.3070		0.1912	
Sex female (ref. male)	0.7082	0.5475	1.2940	2.0304	0.1958	
Age	−0.0750	0.0367	−2.0430	0.9277	0.0410	*
Education	0.3912	0.2255	1.7350	1.4788	0.0827	
Self-rated health intermediate (ref.: unsatisfactory)	0.7535	0.6656	1.1320	2.1245	0.2576	
Self-rated health satisfied (ref.: unsatisfactory)	1.5440	0.6260	2.4660	4.6835	0.0137	*
Self-perception compared to others	0.4938	0.1871	2.6390	1.6385	0.0083	**

**Table 3 ijerph-18-11515-t003:** Ordinal model of self-rated health regressing on several conditions in separate models controlling for sex, age, and education (data not shown); * *p* < 0.05, *** *p* < 0.001.

	Estimate	Std. Error	z Values	*p*	
Memory problems yes (ref: no)	−0.5606	0.3380	−1.6590	0.0972	
Walking problems yes (ref: no)	−3.1927	0.4903	−6.5120	0.0000	***
Debility of eyesight yes (ref: no)	−0.1852	0.3321	−0.5580	0.5772	
Hardness of hearing yes (ref: no)	−0.0501	0.3449	−0.1450	0.8846	
Vertigo yes (ref: no)	−1.7193	0.3625	−4.7430	0.0000	***
Balance problems yes (ref: no)	−1.4844	0.3781	−3.9260	0.0001	***
Orientation problems yes (ref: no)	−1.0608	0.4824	−2.1990	0.0279	*

**Table 4 ijerph-18-11515-t004:** General linear model of self-perception compared to others regressing on several conditions in separate models controlling for sex, age, and education (data not shown); * *p* < 0.05, ** *p* < 0.01, *** *p* < 0.001.

	Estimate	Std. Error	z Values	*p*	
Memory problems yes (ref: no)	−0.3954	0.2226	−1.7770	0.0777	
Walking problems yes (ref: no)	−1.1458	0.2084	−5.4970	0.0000	***
Debility of eyesight yes (ref: no)	−0.2900	0.2253	−1.2870	0.2001	
Hardness of hearing yes (ref: no)	−0.4058	0.2273	−1.7850	0.0764	
Vertigo yes (ref: no)	−0.6778	0.2526	−2.6830	0.0082	**
Balance problems yes (ref: no)	−1.1973	0.2605	−4.5970	0.0000	***
Orientation problems yes (ref: no)	−0.7179	0.3530	−2.0340	0.0439	*

**Table 5 ijerph-18-11515-t005:** Tetrachoric correlation matrix of the four “significant conditions”.

	Walking Problems Yes (Ref: No)	Vertigo Yes (Ref: No)	Balance Problems Yes (Ref: No)	Orientation Problems Yes (Ref: No)
Walking problems yes (ref: no)	1	0.507	0.592	0.352
Vertigo yes (ref: no)		1	0.86	0.5103
Balance problems yes (ref: no)			1	0.6285
Orientation problems yes (ref: no)				1

## Data Availability

Not applicable.

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
