# Peer review of "Let’s Walk It: Mobility and the Perceived Quality of Life in Older Adults"

_ijerph, 2021, doi:10.3390/ijerph182111515_

Round 1

Reviewer 1 Report

The abstract could be improved by providing a greater rationale for conducting the study. There is no mention of why this is an important study and how it is adding to knowledge. 

Mentioning the mean age in the abstract is recommneded, given the wide range of ages in the study. 

The manuscript requires extensive editing by a native English speaker. 

The introduction needs to be improved immensely. There is no meniton of quality of life and the factors that may influence self perceived QOL. The authors could also talk about movement and exericse and how this influences QOL. 

The introduction does not state any hypotheses, aims and gap in knowledge that the current study is addressing. 

Lines 39-48 sound like a methods section instead of an introduction. 

Was the questionnaire that was used to measure QOL valid and reliable? 

The methods section needs more clarity. Recruitment is not discussed, and the order of assessments is not clear. 

The results need to be displayed more clearly. 

There is a lack of discussion regarding the health status of participants. This could influence both mobility and QOL. Other variables such as nutrition/diet and medication intake are not discussed. 

There is no mention of the mechanisms linking mobility to QOL. Longitudinal, prospective trials are needed and randomised controlled trials to better determine the examined link. 

Vertigo is likely to be associated with a host of co-morbitities that impact qiality of life, making it difficult to determine whether this relationship is caused by vertigo alone. 

Reviewer 2 Report

This is an interesting study examining the relationship between mobility and quality of life in older adults.

I have major concerns regarding this manuscript and I don't think it is really for publication in the current state.

  1. First, the current literature review is very narrow and limited. The main weakness of the paper occurs from its theoretical frame. It is hard to understand what is the paper main contribution to the literature. The introduction should be focused on the gaps of literature and how this paper fill this gap
  2. Each mediating variable, such as self-rated health and self-perception, should be discussed better in the introduction and link to theoretical framework
  3. Each of the analysis that is conducted should be explained better. Each analysis should be linked to the research question that the authors aimed to answer
  4. The validity and reliability of the measures should be reported
  5. The results section should be restructured better to increase the readability

Round 2

Reviewer 1 Report

You have made substantial changes to the manusript. The abstract, introduction and methods have been improved. All the best with your future research. 

Reviewer 2 Report

The authors have sufficiently addressed my comments